# Aya in Action: An Investigation of its Abilities in Aspect-Based Sentiment Analysis, Hate Speech Detection, Irony Detection, and Question-Answering

## Abstract

While resource-rich languages drive considerable advancements, low-resource languages face challenges due to the scarcity of substantial digital and annotated linguistic resources. Within this context, in 2024, Aya was introduced, a multilingual generative language model supporting 101 languages, over half of which are lower-resourced. This study aims to assess Aya's performance in tasks such as Aspect-Based Sentiment Analysis, Hate Speech Detection, Irony Detection, and Question-Answering. Our methodology consists of utilizing a few-shot learning approach, incorporating examples from the ABSAPT 2022, ToLD-BR, IDPT 2021, and SQUAD v1.1 datasets as prompts for inference. The objective is to evaluate Aya's effectiveness in these tasks without fine-tuning the pre-trained model, thereby exploring its potential to improve the quality and accuracy of outputs in various natural language understanding tasks. Results indicate that while Aya performs well in certain tasks like QA, where it surpassed Portuguese-specific models with a 58.79% Exact Match score, it struggles in others. For the Hate Speech Detection task, Aya's F1-score of 0.64 was significantly lower than the 0.94 achieved by the Sabiá-7B model. Additionally, the model's performance on the ABSA task improved considerably when neutral examples were excluded, but its handling of complex slang and context-dependent features in other tasks remained challenging. These results suggest that multilingual models like Aya can perform competitively in some contexts but may require further tuning to match the effectiveness of models specifically trained for Portuguese.

## 1 Introduction

In recent years, advances in Large Language Models (LLMs) have predominantly focused on a narrow set of data-rich languages, leaving aside a vast number of languages with fewer resources available (Nguyen et al., 2023). Brazilian Portuguese, considered a low-resource language, falls into this context and therefore encounters limited resources available for the development of a model that comprehends the nuances of the Brazilian language.

This disparity demonstrates a problem within the Natural Language Processing (NLP) domain, where resource-rich languages, such as English, lead significant advances, while many other low-resource languages lag behind (Held et al., 2023; Sengupta et al., 2023). The lack of substantial digital and annotated linguistic resources for these languages makes it difficult to create effective linguistic models, which in turn affects numerous applications ranging from machine translation to sentiment analysis and more.

In this context, Üstün et al. (2024) introduced Aya, a multilingual generative language model supporting 101 languages, with more than half of them being low-resource. The authors highlight a critical issue in machine learning: how to effectively capture the nuances of the "long tail" — the rare and underrepresented examples of language that make up much of the real world. Aya represents a significant step forward by addressing the needs of these underrepresented languages, offering a more inclusive solution in NLP by expanding the reach of advanced models beyond high-resource languages.

Given the insufficient quantity of resources for Brazilian Portuguese, this study aims to assess the performance of Aya-101 in a range of NLP tasks specific to Brazilian Portuguese, including Aspect-Based Sentiment Analysis (ABSA), Hate Speech Detection (HS), Irony Detection (ID), and Question-Answering (QA). We employ a few-shot methodology to evaluate the model's effectiveness, as this approach is particularly well-suited for low-resource scenarios where extensive labeled datasets are unavailable. By assessing these tasks, we aim to catalog Aya's performance quality in low-resource contexts, where data limitations pose significant challenges to NLP models. This evaluation will allow us to systematically gauge how well the model adapts to the nuances of Brazilian Portuguese across various tasks.

The paper is organized into the following sections: **Theoretical Background** provides key concepts related to domain knowledge on the strategies used, technical information necessary for understanding the tasks addressed, and a brief overview of LLMs; **Related Works** examines relevant literature, with a particular emphasis on studies involving NLP models for the Portuguese language and other low-resource languages; **Methodology** outlines the experimental procedures, including details about datasets, few-shot strategies, and data flow across tasks; **Experiments** presents the metrics and compares the results; **Final Remarks** summarizes the findings and offers a brief discussion on potential future research.

## 2 THEORETICAL BACKGROUND

In this section, we will cover the basic ideas behind fundamental concepts in NLP. These include Sentiment Analysis (SA), ABSA, HS, ID, QA, LLMs and Few-shot Learning (FSL).

SA is the task that aims to identify opinions expressed towards an entity within various forms of content. There are different levels of granularity that may be applied, each one helping to understand different aspects of opinions. The most common granularity levels are Document-Level, Sentence-Level, and Aspect-Level (also known as Aspect-Based Sentiment Analysis) (Liu, 2015). The Document-level can only obtain an opinion for an entire document, making it unable to handle multiple opinions; Sentence-level is a more complete granularity level, as it can extract multiple opinions from a single document, however, it only extracts those multiple opinions for the entity as a whole, not being able to understand which part of the entity those opinions are aimed towards. Lastly, ABSA is a finer-grained approach that aims to extract, for a given text, exactly which aspects of the target entity are present in the text and their respective polarities. This method enhances the understanding of the positive and negative parts of a product or service. However, the application of ABSA is not limited to these two areas, as it can also be applied in other contexts, such as analyzing the sentiment of a politician's statement. In this research, we will focus the SA evaluation on this granularity level within text content, specifically on the classification of the polarities of predefined aspects in a given text.

HS aims to identify potentially aggressive references to individuals or groups in texts. The "Aggressive" reference may be either some form of hatred, incitement of violence, or other related harmful content, making this task of great significance for social media platforms, which are a key domain for the spread of hateful content. This is a specially difficult task, since the categorization of hate content does not directly relate to profanity; it requires to be targeted to some individuals or groups (Mondal et al., 2017). Furthermore, multiple words may be considered hate speech depending on the context in which they are used, while in other circumstances, they are simply ordinary words without any aggressive or hateful meaning.

ID is concerned with detecting an ironic meaning in texts. This is a challenging task that can be used in multiple contexts, as the identification of irony in a text can completely change its interpretation, thereby changing any understanding of that text, that may have been obtained with different analysis. The main challenge associated with ID is the complexity of the ironic behaviour, which heavily relies on context, that may not necessarily be present in the processed text, and may depend on prior knowledge about the opinions of the speaker.

QA can be divided into three main components: question classification, information retrieval, and answer extraction (Allam & Haggag, 2012). In the question classification stage, the objective is to identify the type of answer expected. For example, in the question "What year did Alan Turing publish his paper on the Turing Machine?", the answer should be a specific year. Information retrieval involves gathering results based on the question and its expected answer type. If no relevant data is found, the process may stop. Finally, the answer extraction provides the answer to the initial question.

LLMs are machine learning models trained to understand and possibly generate natural language. These models are usually based on Transformers (Vaswani et al., 2017), and are trained on vast amounts of textual data from different sources, enabling the understanding of the natural language patterns across multiple contexts. While a great number of LLMs exist, they are usually primarily trained for English, which makes them very good at that specific language, but have worse performance on other languages. This disparity occurs due to differences in text, as well as in the cultural and local references that exist in texts from multiple languages, which can not be easily translated considering distinct languages.

Due to the excessive costs of training LLMs, one common approach that allows for the representation of multiple languages without a specialized model is the use of Multilingual Models. These models work the same way as regular specialized LLMs, however they are trained using data from multiple languages at once, making them able to understand the particularities of multiple languages in a single model.

FSL is a machine learning technique that focuses on training models with minimal labeled data, in order to save memory and processing. This method is particularly useful when pre-training is resource-intensive or impractical. The concept of learning from limited experience aligns with the foundational idea of machine learning, as stated by Mitchell (1997) in his work:

> A computer program is said to **learn** from experience $E$ with respect to some class of tasks $T$ and performance measure $P$, if its performance at tasks in $T$, as measured by $P$, improves with experience $E$. (Mitchell, 1997).

This definition encapsulates the core principle of FSL, where the model must generalize from minimal examples to improve performance on a broader class of tasks.

Aya is a multilingual generative language model introduced by Üstün et al. (2024), supporting 101 languages, with over half being low-resource. This model prioritizes inclusivity by rigorously addressing issues of toxicity, bias, and safety. It enhances performance through fine-tuning and data pruning, outperforming benchmarks like mT0 (Sanh et al., 2022) and BLOOMZ (Workshop et al., 2023) while covering a broader range of languages. Aya represents a significant step towards greater accessibility in language models.

## 3 RELATED WORKS

In this section, we conduct a comprehensive analysis within the scope of NLP, focusing on literature research concerning ABSA, HS, ID, and QA. We examine significant contributions and the methods used in each area, highlighting key findings and progress that have enhanced our understanding and capabilities in these fields. The primary criterion was the relevance of the research to specific NLP tasks in Portuguese, particularly in the domains of the tasks in our study, as well as challenges faced in low-resource languages. Additionally,

the selection included comparative studies that analyzed performance metrics across various models and datasets.

The first shared task dedicated for ABSA in the Portuguese language was proposed by da Silva et al. (2022) at the Aspect-Based Sentiment Analysis in Portuguese (ABSAPT) competition. The competition was divided in two diferent sub-tasks: Aspect Extraction (AE), which focused on extracting aspects of texts, and Aspect Sentiment Classification (ASC), which is the classification of the sentiment for those aspects. In the AE task, the best results were achieved by methods based on Transformers encoder-only models, with an Accuracy (Acc) of up to 0.67 (Gomes et al., 2022). For the ASC task, the best results were obtained using encoder-decoder Transformers models, with an ensemble of four fine-tuned PTT5 (Carmo et al., 2020) models, achieving an Acc score up to 0.82.

Wenxuan, Yue, Liu, Sinno, and Lidong evaluate the use of LLMs in various sub-tasks of SA (Zhang et al., 2024). Their study indicated that for straightforward tasks like document and sentence-level sentiment analysis, LLMs using FSL outperform smaller fine-tuned encoder-decoder models. However, for ABSA and particularly in the ASC task, the results were comparable. Yet, when both tasks are combined, the fine-tuned models significantly outperform the few-shot LLMs.

Leite et al. (2020b) used a data-driven approach based on the ToLD-BR (Toxic Language Dataset for Brazilian Portuguese) dataset (Leite et al., 2020a). They divided the dataset into standard training, development, and test sets, utilizing a Bag-of-Words representation and AutoML for their initial model (BoW + AutoML). By employing the auto-sklearn and simple transformers libraries, they optimized the process with default parameter tuning and ensured consistency with a fixed seed. Their evaluation of two BERT models, mBERT (Devlin et al., 2018) and BERTimbau base (Souza et al., 2020), showed promising outcomes, achieving F1-Scores (F1) of 0.75 and 0.76, respectively, which surpassed the BoW + AutoML baseline.

Regarding the ID task, Corrêa et al. (2021) introduced the first shared task focusing on detecting irony in Portuguese texts, tweets and news articles at the Irony Detection in Portuguese (IDPT) competition in 2021 (Corrêa et al., 2021). Their findings revealed that classical feature-based models outperformed deep learning approaches on the IDPT 2021 tweets dataset, achieving a Balanced Accuracy (BAcc) of 0.52. Addressing this issue, Jiang et al. (2021) proposed a solution using BERTimbau (Souza et al., 2020), weighted loss functions, and ensemble learning. Jiang demonstrated that the most effective approach involved leveraging two datasets from the IDPT 2021 for model training and generalization, achieving a BAcc of 0.48. Given the limited size of the IDPT 2021 dataset (Subies, 2021), Jiang opted to employ Data Augmentation techniques. This involved randomly masking 15% of tokens and utilizing BERTimbau base with hyperparameter Grid Search to predict the masked tokens, resulting in a BAcc of 0.49 in experiments with BERTimbau.

In 2023, Aytekin & Erdem (2023) evaluated the use of Generative Pre-trained Transformer (GPT) models for the task of ID in English, using Zero Shot Learning (ZSL) and FSL examples. Their study focused on the text-davinci-003 and gpt-3.5-turbo models (Radford et al., 2019), employing a FSL pproach. The models demonstrated their effectiveness in ID, achieving the highest F1 among the models tested, and the best Recall (R) in a binary classification task compared to others in the competition.

One significant challenge in QA is the limited availability of high-quality datasets, particularly in languages other than English. Less-resourced languages, such as Brazilian Portuguese, often lack comprehensive QA datasets, making it difficult for researchers to explore and evaluate the latest techniques in QA.

The work proposed by Bahak et al. (2023) analyzed ChatGPT's (Achiam et al., 2023) role as a Question Answering System (QAS) and compared it with other QASs. The study primarily evaluated ChatGPT's ability to extract answers from provided paragraphs, a core QAS function. It also examined performance without contextual passages. Several experiments on response hallucination and question complexity were conducted using established QA datasets, including SQUAD v1.1 (Rajpurkar et al., 2016), NewsQA (Trischler et al., 2017), and Persian-QuAD (Kazemi et al., 2022), in both English and Persian. The research indi-

cates that ChatGPT lags behind task-specific models in QA effectiveness. It demonstrates that providing context and utilizing prompt engineering can enhance performance, particularly for questions without explicit answers in the text. Notably, the results comparing effectiveness between various Language Models on SQuAD 1.1, show that ChatGPT presented the worst Exact Match (EM) score of 44.4, between LUKE (Yamada et al., 2020) (90.2), XLNet (Yang et al., 2019) (89.9), and SpanBERT (Joshi et al., 2020) (88.8).

Nunes et al. (2023) proposed a study using LLMs for high-stakes multiple-choice tests in Brazilian Portuguese, directly addressing the challenges of limited QA datasets. Their research helps overcome the scarcity of extensive QA datasets in languages like Brazilian Portuguese by utilizing advanced models, particularly GPT-4 (Achiam et al., 2023) with Chain-of-Thought prompts. The performance and accuracy of GPT-4 on questions from the Exame Nacional do Ensino Médio (ENEM), a major entrance exam for Brazilian universities, are impressive and show the potential of LLMs in tackling complex QA tasks in Portuguese.

Ram et al. (2021) demonstrated the challenges posed by the FSL setting in QA benchmarks, where only a few hundred training examples are available. The authors noted that the standard models struggle in this scenario, showing a gap between common pre-training objectives and the needs of QA tasks. To address this, they proposed a novel approach for returning answers. In this method, they masked all but one recurring span within each set in a passage. This approach showed promising results, with the model achieving a remarkable 72.7% F1 on the English version of SQUAD v1.1 (Rajpurkar et al., 2016) using only 128 training examples.

The research of *Removed for Anonymous Review* evaluated the performance of BERTimbau Base and Large models across various NLP tasks, including SA, AE, HS and ID tasks. The study consisted in fine-tuning the models, applying them for tasks, testing the models over the datasets of TweetSentBR (Brum & Nunes, 2018), ABSAPT 2022 (da Silva et al., 2022), ToLD-BR (Leite et al., 2020b) and IDPT2021 (Corrêa et al., 2021), and evaluating the results using six metrics: Accuracy (Acc), Precision (P), Recall (R), F1-Score (F1), Specificity (S) and Balanced Accuracy (BAcc). The results achieved are also displayed in Table 2.

Additionally, *Removed for Anonymous Review* presented results for AE, SA, HS, ID and QA tasks using the Albertina PT-BR Large and Base models, comparing them to the their previous work which used BERTimbau Base and Large models. Following a series of fine-tuning and testing phases on the same four datasets, Albertina PT-BR models showed promising results, with performance varying across tasks. ID exhibited the most significant improvements, with Albertina PT-BR achieving slightly lower Acc only in the Base version models, at 41% compared to 40%. QA also demonstrated enhancements, evaluated using metrics such as F1 and EM. These findings contribute to the practical application and evaluation of the Albertina PT-BR model, particularly in the context of Brazilian Portuguese.

The Sabiá-7B model (Pires et al., 2023), specialized in Brazilian Portuguese, also underwent evaluation across multiple NLP tasks, including ABSA, HS, QA and ID (*Removed for Anonymous Review*), and then compared to the two previously mentioned works. Similarly, this study used the same four datasets (Brum & Nunes, 2018; da Silva et al., 2022; Leite et al., 2020b; Corrêa et al., 2021) with the few-shot approach and prompt engineering techniques. The results demonstrated that Sabiá-7B achieved impressive performance, with a particular emphasis on the HS task. However, it showed some limitations in the QA task, where it struggled to generate the precise answers required for the Exact Match metric. The results of this and the previous two works are compared in table 2.

## 4 METHODOLOGY

Our approach involves three primary phases. Initially, we pre-process the few-shot examples by selecting and organizing samples from the datasets (ABSAPT 2022 (da Silva et al., 2022), ToLD-BR (Leite et al., 2020b), IDPT 2021 (Corrêa et al., 2021), and SQUAD v1.1 (Rajpurkar et al., 2016)). This step involves curating relevant data for each task to form a "training set", ensuring that the examples are representative

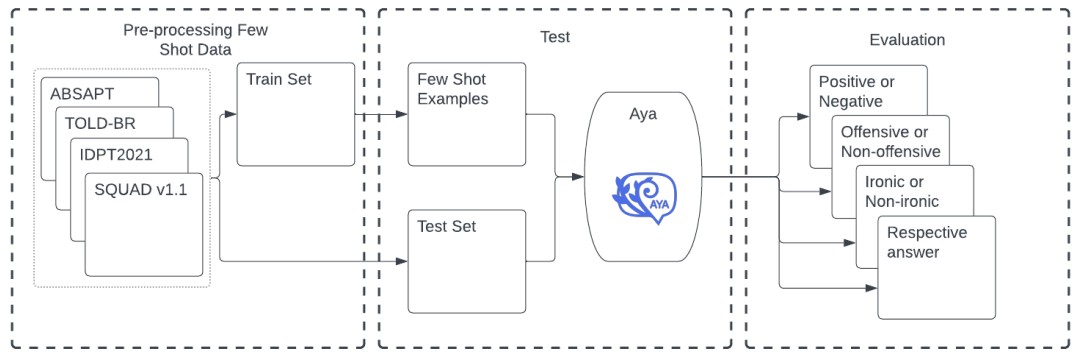

Figure 1: Methodology of this work.

and diverse. The examples used as few-shot are taken from the original training set, but only a few examples were used, as their use is limited by the context length of the model.

So, given this restriction, we selected the maximum number of examples that could fit with all test examples in the context length, selecting them to be as balanced as possible. For the HS and ID tasks, this means that half were positive and half negative, and for the ABSA task 4 were positive, 4 negative, and 3 neutral, all of them containing different aspects. For the QA task, we selected only four examples, in order not to exceed the model's input token limit. For a balanced division, we chose diverse questions that represented the scope of the dataset in a more general way, using questions that began with "What", "Where", "Who" and "When".

This choice was informed by the most frequent question starters observed in the dataset, which includes terms like "qual" (what) with 15174 occurrences, "o" (the) with 12713 occurrences, "que" (what or which) with 8844 occurrences, "quem" (who) with 7969 occurrences, "em" (in or on, depending on context) with 7083 occurrences, and "quando" (when) with 5453 occurrences. These terms not only appear most frequently in the dataset but also reflect common interrogative forms in Brazilian Portuguese. By focusing on these question types, we ensure that the selected examples included a wide range of question categories, such as identifying objects, locations, persons, and temporal information.

Next, we incorporate the few-shot examples as prompts for each inference, serving as input to the model, which is the previously introduced Aya-101 (Üstün et al., 2024). Finally, we analyze the results obtained in each task. This analysis involves evaluating the model's performance, comparing it to baseline results, and assessing the effectiveness of using few-shot examples. Through this process, we aim to understand the strengths and limitations of our approach and identify areas for further improvement.

Table 1: The division used for each dataset.

| Sets | ABSA | HS | ID | QA |
|---|---|---|---|---|
| Original Train | 3,111 | 16,750 | 15,211 | 87,599 |
| Original Test | 686 | 2,094 | 300 | 10,570 |
| Few-Shot Examples | 11 | 10 | 20 | 4 |
| Test Set | 686 | 150 | 300 | 4,139 |

In the first step, we adopt a FSL approach for each task to generate predictions. To create the few-shot examples, we selectively picked instances from the dataset and include them alongside each test example during inference. The selection process prioritizes diversity in the examples, aiming to cover a wide range of cases while ensuring that the total number of tokens remains within the model's maximum context length. After inference, we analyze the model's output, which can be a label (for ABSA, HS, and ID tasks) or an answer (for QA tasks).

In the ABSA task, we selected eleven few-shot examples that are balanced in terms of the aspects they cover and their polarities. The examples contain nine different aspects, including four examples with negative polarity, four with positive polarity, and three that are neutral. Each few-shot example is formatted as "Text: REVIEW_TEXT Aspect: ASPECT Sentiment: POLARITY", with the capitalized fields replaced by their respective values from the dataset. The final prompt for each text to be predicted include a base prompt and the eleven few-shot, followed by the example with the same structure, but with the "POLARITY" field removed, requiring the model to predict only the sentiment label. The base prompt is in Portuguese and says *"You must classify the sentiment of the given aspect in the following texts. Each sentiment should be labeled as 'Positive', 'Neutral', or 'Negative'. Consider only the sentiment for the specified 'Aspect' in each text"*. The total number of texts inferred is 686, that is the ABSAPT shared task total test set.

For the HS task, ten texts were selected from the ToLD-BR dataset as few-shot examples: five labeled as hate speech and the other five as non-hate speech. The ToLD-BR is a dataset that contains toxic speech, which considers hate speech, offensive speech, and aggressive speech as the same category. The examples are formatted as "Text: EXAMPLE_TEXT Label: LABEL", and the example to be predicted follows the same format, but without the "LABEL", which the model must generate. Note that for this task, we did not use a base prompt. After some experimentation, we noticed that the generation was more efficient when using labels as numbers, instead of actual labels, so all labels were changed to '1' or '0'. The test set contains 150 examples, half of them containing hate speech and half not containing it.

In the ID task, twenty few-shot examples were used, being half of them ironic, and half non-ironic. The format was the same as to the HS task, with labels also changed to '1' and '0' as well. For testing, a total of 300 texts were inferred, which represents the complete IDPT 2021 test set. The following translation was used as base prompt: *"Classify, as in the examples below, whether the text excerpts are ironic (POSITIVE) or not ironic (NEGATIVE)"*.

The methodology for the QA task includes not only four few-shot examples, but also specific instructions. For this task, the following prompt was included: *"The answer to each question is a segment of text from the corresponding reading passage. The answer should be extension based, objective answer only. Answer the question accurately and succinctly, containing only your main answer, as short as possible, as in the examples below:"*. This prompt serves as the instructions added before the few-shot examples. We used a combination of the base prompt and the few-shot examples as prompt for the ABSA, ID, and QA tasks, while the HS task did not require a base prompt.

All examples consist of a context, a question, and the expected answer. This structure requires more tokens per example than in the other tasks, so we are limited to including only four examples as few-shot, while ensuring enough spare tokens in the context length for each test example. Given this limitation, we had to carefully select the examples to be used for the few-shot, so they can include the types of question contained in the dataset. Therefore, each of the four examples covers a distinct type of question: one for each of "What?", "Where?", "Who?" and "When".

For the evaluation of this approach, we used a total of 4139 examples from the test set portion of the SQUAD v1.1 dataset. Each one of those examples included, along with the instructions prompt and the few-shot examples, the context and question of the example, and the model was tasked to generate the answer to that specific question. Then, we compared only the generated answer with the expected one.

## 5 EXPERIMENTS

The Aya-101 model was tested on four tasks: ABSA, ID, HS, and QA. Each test dataset was evaluated on several metrics, such as Accuracy (Acc), Precision (P), Recall (R), and F1-Score (F1) (Brownlee, 2016), except for the QA task, which was evaluated based on Exact Match (EM) and F1 only, and the ABSA task, which was evaluated also on the Balanced Accuracy (BAcc).

The components of the equations are based on different types of predictions that the model can make. In this sense, when a model produces a *True Positive* result, it correctly produces a positive value for the task. For example, if the task is to identify hate speech, a *True Positive* occurs when the model correctly classifies the content as hate speech. Similarly, a *False Positive* means the model erroneously predicts a positive result; in this case, the classification is incorrect and, using the example, a non-hate speech content is misclassified as hate speech. A *False Negative*, on the other hand, refers to cases where the model fails to identify the positive outcome, resulting in an erroneous negative result. Finally, *True Negative* means the content is non-hate speech and is correctly classified as such by the model.

In the context of the Balanced Accuracy equation, the *Recall Pos* refers to the recall for the positive class, the *Recall Neg* to the negative class, and the *Recall Neu*, to the neutral class. Each value is calculated by comparing the correctly identified instances of the class against the false predictions, ensuring that the model's performance across all classes is equally weighted. Furthermore, the *Total Number of Instances* represents the total number of samples analysed by the model.

$$Accuracy = \frac{True\ Positives + True\ Negatives}{Total\ Number\ of\ Instances} \tag{1}$$

$$Precision = \frac{True\ Positives}{True\ Positives + True\ Negatives} \tag{2}$$

$$Recall = \frac{True\ Positives}{True\ Positives + False\ Negatives} \tag{3}$$

$$F1 - Score = 2 * \frac{Precision.Recall}{Precision + Recall} \tag{4}$$

$$BAcc = \frac{(Recall_{Pos} + Recall_{Neu} + Recall_{Neg})}{3} \tag{5}$$

$$ExactMatch = \frac{TruePositives}{TotalNumberofInstances} * 100 \tag{6}$$

In Table 2, we show the results of previous works (*Removed for Anonymous Review*, *Removed for Anonymous Review*, *Removed for Anonymous Review*) . The methodology in these studies also employed a FSL approach, incorporating examples from the same datasets utilized in this research, but using different Transformers models trained for the Portuguese language (BERTimbau, Albertina PT-BR, and Sabiá-7B). They applied this procedure to NLP tasks: HS, ID and QA.

For the ABSA task, we present two sets of results: in the first, we show the metrics obtained for the complete test set, and in the second, we show the results of the predictions after removing the examples with a "neutral" target polarity. This is done to clearly show the difficulties of the approach. The main metric for this task is the BAcc.

In the first set, with all examples, the BAcc obtained is of only 0.61, which is worse than all teams that submitted results to ABSAPT. This value is mainly due to the model's inability to handle "neutral" polarities,

Table 2: Results Obtained Using BERTimbau, Albertina PT-BR, Sabiá-7B and Aya models. In this research, we explore the results of Aya, while for all other models, there is related prior work.

| Model | Task | Dataset | Acc | P | R | F1 | BAcc | EM% |
|---|---|---|---|---|---|---|---|---|
| BERTimbau Base | HS | ToLD-BR | 0.88 | 0.89 | 0.88 | 0.88 | - | - |
| | ID | IDPT 2021 | 0.41 | 0.36 | 0.41 | 0.25 | - | - |
| | QA | SQUAD v1-PT | - | - | - | 0.56 | - | 43.29 |
| BERTimbau Large | HS | ToLD-BR | 0.89 | 0.90 | 0.89 | 0.89 | - | - |
| | ID | IDPT 2021 | 0.40 | 0.16 | 0.40 | 0.22 | - | - |
| | QA | SQUAD v1-PT | - | - | - | 0.62 | - | 47.15 |
| Albertina Base | HS | ToLD-BR | 0.78 | 0.72 | 0.77 | 0.74 | - | - |
| | ID | IDPT 2021 | 0.40 | 0.40 | 0.99 | 0.57 | - | - |
| | QA | SQUAD v1-PT | - | - | - | 0.57 | - | 45.12 |
| Albertina Large | HS | ToLD-BR | 0.58 | 0.34 | 0.58 | 0.43 | - | - |
| | ID | IDPT 2021 | 0.41 | 0.41 | **1.0** | **0.58** | - | - |
| | QA | SQUAD v1-PT | - | - | - | 0.32 | - | 47.30 |
| Sabiá-7B | ABSA | ABSAPT 2022 | 0.77 | 0.64 | 0.61 | 0.53 | 0.61 | - |
| | ABSA* | ABSAPT 2022 | - | - | - | **0.79** | **0.91** | - |
| | HS | ToLD-BR | **0.94** | **0.92** | **0.94** | **0.93** | **0.94** | - |
| | ID | IDPT 2021 | 0.46 | 0.50 | 0.46 | 0.44 | - | - |
| | QA | SQUAD v1-PT | - | - | - | 0.54 | - | 39.17 |
| Aya-101 | ABSA | ABSAPT 2022 | 0.77 | 0.51 | 0.46 | 0.43 | 0.61 | - |
| | ABSA* | ABSAPT 2022 | - | - | - | 0.78 | 0.88 | - |
| | HS | ToLD-BR | 0.68 | 0.72 | 0.65 | 0.64 | 0.65 | - |
| | ID | IDPT 2021 | **0.50** | **0.66** | 0.50 | 0.44 | - | - |
| | QA | SQUAD v1-PT | - | - | - | **0.76** | - | **58.79** |

as none "neutral" prediction was generated. For instance, in the example "The hotel is right in the center and is great during the day because it is close to shops, restaurants, pharmacies, the municipal market and the street market. But at night it is a desert, and there is no way to go out alone. The rooms have wooden carpets and are very small. In some rooms, you have to go in first or bring your suitcase. The shower cubicle is also tiny. The breakfast is very good and the cleanliness and bedding are very good.", where the target aspect "hotel" has a neutral polarity, the model incorrectly predicted it as positive. When we exclude the neutral examples, marked with a "*" in Table 2, the results go up to 0.88, a higher value than that achieved by any of the ABSAPT participants, although Sabiá continues to have the highest BAcc, at 0.91.

However, it's important to note that ABSAPT Competition results include neutral examples, which are usually harder than negative and positive examples, as ambiguity is more commonly found in the "neutral" examples than in the positive or negative ones.

Comparing these results with those obtained with the Sabiá-7B model, the Aya-101 model showed overall very similar results, but with a slight increase in the prediction of positive examples, with 92.36% of them correctly predicted (Sabiá-7B correctly predicted 90.22%), and a bigger decrease on negative examples, with 85.38% compared to Sabiá's 92.31%. For the neutral examples, the predictions were approximately split in

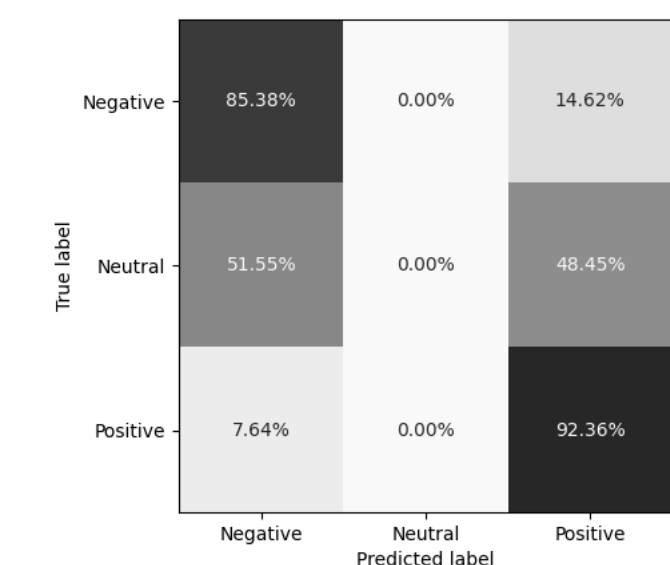

Figure 2: Confusion matrix of the results in the ABSA task.

half between Positive and Negative, showing that the model didn't had a strong bias towards one of them, and was only incapable of handling the neutrals, as shown in Figure 2.

In the HS task, Aya's results were lower than almost all other models, with an F1 of 0.64, significantly worse than the best result, from the Sabiá-7B model, which got a F1 of 0.94. One possible explanation is that the ToLD-BR dataset contains texts with many Brazilian slang words, which are usually the specific word that defines the hateful content of the text. Taking this into account, it's possible that the multilingual training approach taken by Aya can not correctly represent those words, thereby losing the contextual meaning of the texts, and being wrong with more frequency. In comparison, the Portuguese focused training of the Sabiá-7B model may better understand the nuances of these words, being able to distinguish hate speech from texts that include similar words, but can't be categorized as some kind of target hate. Given that non-hateful texts can also include the same slang, since the used texts come from an informal context, it's needed for the model to differentiate between what is targeted hate, and what is just used to emphasize what's being said, or used in a casual manner.

The results obtained for the ID task were worse than those of the Albertina models, higher than those of the Bertimbau models, and close to the Sabiá model. All models produced unsatisfactory results, with less than 0.5 F1 for almost all of them. This happens mainly due to the complexity of the ID task, which depends on a deep understanding of the texts that are being evaluated, and may even require some context from things that are not present in the texts (such as ironic references to news). Since none of the models have access to the external context, the simple understanding of the language may not be enough for the correct classification of the examples.

In the QA task, the Aya model obtained significantly better results than other models. The EM rate was 58.79%, indicating the percentage of questions that were answered perfectly (i.e., it managed to generate an answer that is exactly equal to the ground truth of the dataset). This indicates that, even without a fine-tuning

approach, this model can better summarize the answers to meet expectations, as this summarization is the main problem found for the Sabiá model, which often generated more contextual information in answers than expected.

One important aspect to notice is that the SQUAD v1-PT dataset used in this study automatically translated from an English dataset. As a result, the examples often contain words that are not translated to Portuguese, such as in "Quem ganhou o MVP para o Super Bowl?" ("Who won the MVP for the Super Bowl?"), where "MVP" is an abbreviation for Most Valuable Player, and monolingual models may experience difficulties translating acronyms to Portuguese. The Aya model's training includes multiple languages, such as English, while BERTimbau, Albertina and Sabiá models are trained solely on Portuguese datasets. This multilingual training may positively affect the results. Additionally, Aya is trained using native and translated datasets, which may further improve the results. It is important to consider that automatic translations may include biases to the way in which the texts are written (Vanmassenhove et al., 2021), that may be present in both the Aya model and the SQUAD v1-PT dataset, but not on models trained without automatically translated texts.

# 6 FINAL REMARKS

In this research, we evaluated the Aya model performance across multiple NLP tasks, specifically ABSA, HS, ID, and QA, with a focus on the Portuguese language, using a FSL approach. The model's results were compared with other Transformers models trained completely for Brazilian Portuguese, in a effort to understand where multilingual models can surpass the native models, and where they are not enough.

In conclusion, our work indicate that the Aya model can efficiently handle the ABSA, HS, ID and QA tasks in Portuguese. However, when compared to other models, the performance appears to be more related to the type of data that used in training and in the tasks, with models trained purely for Portuguese obtain better results on datasets that contain native texts, while the multilingual Aya model outperforms them on an automatically translated dataset. Also, the presence of hard tasks, such as ID, indicate that a few-shot learning approach may not be enough for the correct classification in some tasks, which may require more advanced efforts to correctly tackle those problems.

Regarding future works, this study allows for a understanding of general aspects that may affect the results, but more research is required in a deeper personalized approach for each task. For each domain, better data selection (and the inclusion of extra data sources that can be helpful to tackle the noticed shortcomings), more focused prompt engineering, and also other approaches, such as fine-tuning for the generative models, can be used to further understand and improve the results in each task.

Also, the use of a greater variety of datasets, including translated and native datasets for all tasks, may be helpful to understand how much the multilingual training impacts the performance, when compared to a language specific training. The impact of the automatic translation on each model is another topic that may be further explored, to understand how much it affects the results.

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

# A APPENDIX

In this section, we provide additional tables that support the main content of the study. These tables contain detailed information on various aspects of the data used for the few-shot method for all the tasks mentioned in this work.

Table 3: Few-shot examples used in the ABSA task from the ABSAPT 2022 dataset.

| id | review | polarity | aspect | start_position | end_position |
|---|---|---|---|---|---|
| 11 | Um bom local para se hospedar ,ótima localização, bem no centro de Porto Alegre.Deixa a desejar no room service, pouca variedade e muito lento o atendimento.Recepção atenciosa, mas um pouco lenta. Café da manhã simples, mas agradável. | 1 | localização | 37 | 48 |
| 331 | Se você quer apenas um local confortável, sem luxo excessivo, limpo, perto da Strip, com uma piscina bacana e ótimo custo, este é o local. Não tem café da manhã incluso, e o breakfast é caro e limitado. Os quartos têm pia e geladeira, então a dica é você comprar seus ingredientes para o café da manhã no supermercado e levar para o hotel. | 1 | quarto | 206 | 212 |
| | | | | Continues on the next page | |

| id | review | polarity | aspect | start_position | end_position |
|---|---|---|---|---|---|
| 407 | A localização é boa, assim como o tamanho e valor dos apartamentos. Ele fica próximo a supermercados e metrô, o que facilita muito. Contudo, os recepcionistas (homens) deixam a desejar (são um pouco rudes). O banheiro tem um cheiro insuportável de urina, agravado pelo fato da limpeza não ser realizada todos os dias. Tivemos que comprar desinfetante para colocar nos vasos. Entretanto, no geral classifico o hotel como bom. | -1 | limpeza | 277 | 284 |
| 709 | É uma boa relação custo-benefício ficar no Juliz. Os quartos não são dos melhores, mas dá para ter uma razoável noite de sono, ainda mais para quem, como eu, ficou somente uma diária. Ponto positivo para a rede Wi-Fi, que funciona perfeitamente.A recepção fechar a noite é um ponto negativo. | -1 | recepção | 247 | 255 |
| 960 | Se você quer apenas um local confortável, sem luxo excessivo, limpo, perto da Strip, com uma piscina bacana e ótimo custo, este é o local. Não tem café da manhã incluso, e o breakfast é caro e limitado. Os quartos têm pia e geladeira, então a dica é você comprar seus ingredientes para o café da manhã no supermercado e levar para o hotel. | 0 | café da manhã | 147 | 160 |
| 1201 | Se você quer apenas um local confortável, sem luxo excessivo, limpo, perto da Strip, com uma piscina bacana e ótimo custo, este é o local. Não tem café da manhã incluso, e o breakfast é caro e limitado. Os quartos têm pia e geladeira, então a dica é você comprar seus ingredientes para o café da manhã no supermercado e levar para o hotel. | 1 | piscina | 93 | 100 |
| | | | | | Continues on the next page |

| id | review | polarity | aspect | start_position | end_position |
|---|---|---|---|---|---|
| 1624 | É uma boa relação custo-benefício ficar no Juliz. Os quartos não são dos melhores, mas dá para ter uma razoável noite de sono, ainda mais para quem, como eu, ficou somente uma diária. Ponto positivo para a rede Wi-Fi, que funciona perfeitamente.A recepção fechar a noite é um ponto negativo. | 1 | custo-benefício | 18 | 33 |
| 1965 | Se você quer apenas um local confortável, sem luxo excessivo, limpo, perto da Strip, com uma piscina bacana e ótimo custo, este é o local. Não tem café da manhã incluso, e o breakfast é caro e limitado. Os quartos têm pia e geladeira, então a dica é você comprar seus ingredientes para o café da manhã no supermercado e levar para o hotel. | 0 | hotel | 333 | 338 |
| 2049 | É uma boa relação custo-benefício ficar no Juliz. Os quartos não são dos melhores, mas dá para ter uma razoável noite de sono, ainda mais para quem, como eu, ficou somente uma diária. Ponto positivo para a rede Wi-Fi, que funciona perfeitamente.A recepção fechar a noite é um ponto negativo. | -1 | quarto | 53 | 59 |
| 2538 | O Hotel tem ótima localização, perto do centro histórico e principais atrações de Porto Alegre.Quarto com ar-condicionado que funcionava bem e cama confortável! Só o banheiro que deixa um pouco a desejar, mesmo assim tudo funcionava muito bem!! O café-da-manhã é bom! Pra quem pretende ficar o dia todo na rua e voltar pro Hotel somente pra dormir, está ótimo!! | 0 | quarto | 95 | 101 |

Table 4: Few-shot examples used in the QA task from the SQUAD v1-PT dataset.

| id | title | context | question | answers |
|---|---|---|---|---|
| 5733be28477 6f41900661181 | University_of_Notre _Dame | Arquitetonicamente, a escola tem um caráter católico. No topo da cúpula de ouro do edifício principal é uma estátua de ouro da Virgem Maria. Imediatamente em frente ao edifício principal e de frente para ele, é uma estátua de cobre de Cristo com os braços erguidos com a lenda "Venite Ad Me Omnes". Ao lado do edifício principal é a Basílica do Sagrado Coração. Imediatamente atrás da basílica é a Gruta, um lugar mariano de oração e reflexão. É uma réplica da gruta em Lourdes, na França, onde a Virgem Maria supostamente apareceu a Santa Bernadette Soubirous em 1858. No final da unidade principal (e em uma linha direta que liga através de 3 estátuas e da Cúpula de Ouro), é um estátua de pedra simples e moderna de Maria. | O que é a gruta de Notre Dame? | {'text': array(['um lugar mariano de oração e reflexão'], dtype=object), 'answer_start': array([415], dtype=int32)} |
| | | | Continues on the next page | |

| id | title | context | question | answers |
|---|---|---|---|---|
| 5733a70c477 6f41900660f62 | University_of_Notre _Dame | Todos os alunos de graduação da Notre Dame fazem parte de uma das cinco faculdades de graduação da escola ou estão no programa do Primeiro Ano de Estudos. O primeiro ano de estudos do programa foi criado em 1962 para orientar calouros em seu primeiro ano na escola antes de terem declarado um major. Cada aluno recebe um orientador acadêmico do programa que os ajuda a escolher classes que lhes dêem exposição a qualquer assunto importante no qual estejam interessados. O programa também inclui um Centro de Recursos de Aprendizagem, que fornece gerenciamento de tempo, aprendizado colaborativo e tutoria de assuntos. Este programa foi reconhecido anteriormente, pelo US News & World Report, como excelente. | Quantas faculdades para alunos de graduação estão em Notre Dame? | {'text': array(['cinco'], dtype=object), 'answer_start': array([66], dtype=int32)} |
| 5733ac31d05 8e614000b5ff6 | University_of_Notre _Dame | O Instituto Joan B. Kroc para Estudos Internacionais da Paz da Universidade de Notre Dame dedica-se à pesquisa, educação e divulgação sobre as causas dos conflitos violentos e as condições para uma paz sustentável. Oferece doutorado, mestrado e graduação em estudos de paz. Foi fundada em 1986 através das doações de Joan B. Kroc, a viúva do proprietário do McDonald's, Ray Kroc. O instituto inspirou-se na visão do reverendo Theodore M. Hesburgh CSC, presidente emérito da Universidade de Notre Dame. O instituto contribuiu para discussões de políticas internacionais sobre práticas de construção da paz. | Qual é o título do Theodore Hesburgh de Notre Dame? | {'text': array(['Presidente Emérito da Universidade de Notre Dame'], dtype=object), 'answer_start': array([0], dtype=int32)} |
| | | Continues on the next page | | |

| id | title | context | question | answers |
|---|---|---|---|---|
| 5733b534477 6f419006610dd | University_of_Notre _Dame | A partir de 2012 [atualização], a pesquisa continuou em muitos campos. O presidente da universidade, John Jenkins, descreveu sua esperança de que a Notre Dame se tornasse "uma das instituições de pesquisa pré-eminentes do mundo" em seu discurso de posse. A universidade tem muitos institutos multidisciplinares dedicados à pesquisa em diversos campos, incluindo o Instituto Medieval, o Instituto Kellogg de Estudos Internacionais, o Instituto Kroc para Estudos Internacionais da Paz eo Centro para Preocupações Sociais. Pesquisas recentes incluem trabalhos sobre conflito familiar e desenvolvimento infantil, mapeamento do genoma, o crescente déficit comercial dos Estados Unidos com a China, estudos em mecânica dos fluidos, ciência e engenharia computacional e tendências de marketing na Internet. A partir de 2013, a universidade abriga o Índice de Adaptação Global Notre Dame, que classifica os países anualmente com base em quão vulneráveis eles são às mudanças climáticas e como estão preparados para se adaptar. | Quem foi o presidente da Notre Dame em 2012? | {'text': array(['John Jenkins'], dtype=object), 'answer_start': array([101], dtype=int32)} |

Table 5: Few-shot examples used in the ID task from the IDPT 2022 dataset.

| id | text | prediction |
|---|---|---|
| 415 | Que pena que eu me esqueci de trazer as folhas de biologia! Agora não posso estudar | 1 |
| 837 | Juro do cartão cai 68 pontos após nova regra do rotativo, diz BC Economia | 0 |
| 2610 | Mais agente percebe"" "" | 1 |
| | | Continues on the next page |

| id | text | prediction |
|---|---|---|
| 2695 | Economia Marcelo felicitou António Costa e Passos Coelho Em | 0 |
| 4233 | estamos entregando a nossa soberania a mestiços"" "" | 1 |
| 4301 | Você guarda dinheiro na poupança? economia Brasil SPC CDL conta Brasil | 0 |
| 5027 | ECONOMIA: Balança comercial brasileira registra superávit recorde em maio | 0 |
| 5401 | E esse é o charme que faz da Rubensliga o estadual mais charmoso do Brasil. | 1 |
| 6162 | Kuhanha é nova accionista principal do banco Moza destaque economia | 0 |
| 6875 | Vou estudar pro Enem amanhã cedo... Acho q dá tempo! | 1 |
| 8389 | Economia Governo espera receita de R$ 43 bilhões com fim da Infraero — Via .twitter.comttyF9a9ISh | 0 |
| 9349 | Na boa, no dia em que eu tiver de colocar aquele treco , vai ser o fim da picada... | 1 |
| 11096 | Primeiro prémio da Lotaria Clássica saiu hoje ao número 04.169 finanças economia dinheiro | 0 |
| 11560 | nossa para de imitar o jeito que eu falei pra ti ontem!!!!!!!!!!!!!!!!!!!!!!!!! | 1 |
| 11863 | Noticia Economia Reus fabricará drones a medida | 0 |
| 12533 | Que tal pregar o que o povo quer ouvir e depois dar conta de todas as almas que irem pro inferno por causa da sua pregação? | 1 |
| 12679 | China muda critério da taxa de paridade do yuan ante o dólar economia | 0 |
| 13380 | A net da escola é muito boa | 1 |
| 13502 | Bovespa opera em alta um dia após protestos contra governo Temer economia via | 0 |
| 13810 | Minha meta de vida: usar uma roupa na cor do rolo de papel higiênico. sqn VivianMaiernoMIS... | 1 |

Table 6: Few-shot examples used in the HS task from the ToLD-BR dataset.

| id | text | toxic |
|---|---|---|
| 17052 | namoral eu so passo vergonha tomar no meu cu | 1 |
| 1374 | carol não quer deixa eu trabalhar mane , toda manda msg essa fdp | 1 |
| 8547 | passei mal na aula da auto escola a muie teve a parar no meio da rua pq se n ia cair da moto mds | 0 |
| 17865 | eu odeio quando mando a porra da mensagem e não me responde | 0 |
| 2974 | caralho, tô chorando | 1 |
| 15295 | rt @user vai trabalhar vagabundo! https://t.co/jizl5d2a56 | 1 |
| 1247 | @user mas de fato existe o hacker então, não é mesmo? ou só porque não é russo tá liberado hacker celulares e vazar mensagens? | 0 |
| 10534 | Minha tia falando pra eu traduzir uma mensagem que um gringo tinha mandado pra ela no Facebook e era só o servidor falando pra ela atualizar o aplicativo https://t.co/f85o2vvIcx | 0 |
| 15616 | @user pelotas o cu dele | 1 |
| 861 | @user engraçado que milo tem 44 acho e o pattinson uns 30! https://t.co/5ns4t2ozm3 | 0 |