# OpenReview forum: "Aya in Action: An Investigation of its Abilities in Aspect-Based Sentiment Analysis, Hate Speech Detection, Irony Detection, and Question-Answering"
_ICLR.cc/2025/Conference — Submitted to ICLR 2025_

### Official Review · Reviewer_9d9r · 2024-11-01

**Soundness:** 1
**Presentation:** 2
**Contribution:** 2
**Rating:** 5
**Confidence:** 4

**Summary:**

The paper assesses Aya’s performance in four tasks: Aspect-Based Sentiment Analysis, Hate Speech Detection, Irony Detection, and question answering in Portuguese language. The authors also compared Aya's performance with other language models. However, it is not understandable which one is from prior studies in Table 2. It is also unclear whether the Sabia-7B model is studied in this study or previous study.

**Strengths:**

The paper uses state-of-the-art multilingual LLMs (Aya), that have been known for their capabilities for low-resource languages. Moreover, the study uses four different tasks (SA, HS, ID, and QA) ranging from classification to text generation/QA.

**Weaknesses:**

The main weakness of this paper is written poorly. The related work and the theoretical background sections are too long. The claim about the examples for few-shot learning could be partially correct but not properly correct. The details of prompting are missing in the paper. The performances are not well discussed in the paper.
I believe the paper will be in good shape if the content is trimmed to a short paper rather than a long full paper.

**Questions:**

**Comments:**
1. How do you ensure the examples of few-shot learning are representative and diverse? What is the measure for representative for example do you consider the classes or the related input text?
2. L259-261, Do all the chosen questions begin with "What", “Where”, “Who” and “When” represents the dataset in a general way?
3. L268-L269, "we selectively remove instances from the dataset and include them alongside each test example during inference." -- Why do you remove instances from the dataset? The removed instances belong to which splits are not discussed.
4. In Table 1, few-shot examples are 11 (SA), 10 (HS), 20 (ID), and 4(QA). Do you use the same examples for all inference data?
5. "In total, they mention nine different aspects, including four examples with negative polarity, four with positive polarity, and three that are neutral." -- Who mentioned? I believe there should be a citation.
6. There are nine different aspects in the dataset, does every aspect represent only one sentiment? If not, how does the example represent the other classes that are not selected?
7. Equations 1-6 are well known to the community, the information should be redundant.
8. L421-422, why neutral is harder than positive and negative is not discussed properly.
9. Comparing the results of two classes (Positive and Negative) with three classes (Positive, neutral, and negative) is not a good idea. Given that the model can easily differentiate two classes where prediction of multi-class is harder.
10. Figure 2, the sum of the confusion matrix for columns is not 100.
11. The authors stated that the model predicted exactly the same answer as the ground truth. However, it would be interesting to see some examples of those answers.
12. SQUAD v1 is an old dataset and there is a possibility of adding this dataset to Aya's training data. It would be interesting to see the performance on some other QA datasets.

---

> ### Author Response · Authors · 2024-11-27
>
> **Summary:**
>
> In this research, we explore the results of Aya, while for all other models, related prior work exists. These other studies are specified in the Related Works section, but the citations themselves have been removed for anonymous review. We will make this more clear in the paper.
>
> **Weaknesses:**
>
> Thank you for your feedback. If we were to transform the paper into a short version, we would end up losing many important details for the replicability of the work. We made some edits to the text, taking your other comments into account.
>
> **Questions:**
>
> 1. As presented in the article, we aimed to balance the examples by considering the labels and adhering to the token limit allowed by the model as input. Thus, for ID, for instance, we would have 50% of the examples as ironic and 50% as non-ironic.
>
> 2. We gathered an overview of the dataset's data and will provide it in the text.
>
> 3. The text is ambiguous. We did not actually remove the examples; we simply separated them as if they were "training" and "testing," where the training set was not used for evaluation.
>
> 4. Yes, we use the same examples for all the inference data.
>
> 5. The text was ambiguous, but we have corrected it now. What we meant to say is that “in total, they, the examples, contain nine different aspects, including four examples with negative polarity, four with positive polarity, and three that are neutral”.
>
> 6. Only the most common aspect (“room”) was included with all polarities. For the other aspects, they appear only once, for a single polarity. They are not supposed to be exhaustive, as the model should be able to generalize that with only a few examples.
>
> 7. We found it important to maintain the equations because they are essential for demonstrating the confusion matrix. The equations provide a clear mathematical representation of the metrics derived from the confusion matrix.
>
> 8.  It is more common to find ambiguity in the "neutral" examples than in the positive/negative ones. We have corrected it in the text.
>
> 9. As mentioned in the article, it is not possible to directly compare results that exclude neutral examples with those that include all examples, as we are excluding the most challenging cases. It is also important to note that we are not altering the problem (it remains a multi-class classification task, where a “neutral” prediction would still be considered an error). We are simply selecting a different visualization of the results to better highlight the strengths and weaknesses of the model/methodology.
>
> 10. The confusion matrix is supposed to sum 100 only in each line, not each column. The lines are the True Labels (from the dataset), and the columns are the Predicted Labels. So, for example, a model that predicts everything as negative would have predicted 100% of the negative as being negative, 100% of the neutral as negative, and 100% of the positive as negative, resulting in 300% on the first column (of the predicted negative), while each line would correctly have a sum of 100%.
>
> 11. “In the QA task, the Aya model obtained significantly better results than other models. The EM rate was 58.79\%, indicating the percentage of questions that were answered perfectly (i.e., it managed to generate an answer that is exactly equal to the ground truth of the dataset).” – In this context, when we say that the model predicted exactly the same answer as the ground truth, we are referring to the Exact Match metric, which indicates the rate at which the model provides answers that exactly match the expected responses in the dataset.
>
> 12. As mentioned in our article, the Portuguese language is a low-resourced language, and, therefore, there aren't many QA datasets. We chose to use SQuAD v1 dataset as it is a well-known and widely used dataset. Additionally, we utilized it because it is an automatically translated dataset, allowing us to examine the model’s nuance when dealing with datasets in native and translated languages.
>
> Thank you!

---

> > ### Comment · Reviewer_9d9r · 2024-11-29
> > **Response to authors reply**
> >
> > Thank you for these clarifications and answers. I will increase my score from 3 to 5 as I believe the authors have taken steps to clarify the asked questions.

---

### Official Review · Reviewer_hqW4 · 2024-11-02

**Soundness:** 3
**Presentation:** 3
**Contribution:** 3
**Rating:** 6
**Confidence:** 5

**Summary:**

This study aims to assess the performance of Aya, a multilingual generative model trained on a wide range of low resource languages and a variety of downstream tasks like Aspect-Based Sentiment Analysis, Hate Speech Detection, Irony Detection, and Question-Answering. The objective is to evaluate Aya's effectiveness in these tasks but only on the pre-trained model without any finetuning. This would reveal its potential to improve the quality and accuracy of outputs in various natural language understanding tasks. Instead, this work employs a few-shot methodology to evaluate the model's effectiveness as this approach is particularly better suited in abscence of extensive labelled data in low resource languages. Results indicate that while Aya performs well in certain tasks like Question-Answering for languages like Portuguese, for other tasks like Hate Speech Detection the performances were significantly underwhelming. These results suggest that multilingual models like Aya can perform competitively in some contexts but may require further tuning to match the effectiveness of models specifically trained for Portuguese.

**Strengths:**

This paper addresses an important problem that aims to mitigate technological inequities towards low resource language.
The methodology used in this paper is reasonable and easy to understand. In addition, the paper also makes a thorough inquiry of related research before carrying out their work.
The experiment section provides adequate amount of evaluation to properly assess the performance of the model for a number of language tasks in a low resource language like Portuguese.

**Weaknesses:**

This work is low in novelty despite addressing an important problem.  Training a generative model for a low resource language like Portuguese is certainly important work. While the authors employ few shot learning as a logical workaround for the issue of low training data, their evaluation reveals the relative limitation of this approach after a point. The authors could further look into some technical innovations in this regard to improve performance of the Aya model in portuguese for tasks like Hate Speech or Irony Detection.

**Questions:**

Other than Portuguese have the authors considered evaluating their model for any other low resource language? That would provide a more comprehensive idea of the performance variance of the Aya across languages.

---

> ### Author Response · Authors · 2024-11-27
>
> **Weaknesses:**
>
> Thank you for your constructive feedback. We will take your suggestions into consideration in our future work, aiming to explore technical innovations to improve the performance of the Aya model and other models we apply these NLP tasks.
>
>
> **Questions:**
>
> The group has other works related to these tasks in the Portuguese language, with one of the objectives being to compare Aya's results with those obtained from other models that support the language. We also aim to encourage the study of Portuguese and its exploration in models that support it, to increase the amount of available resources. Thus, we did not consider evaluating this model for other low-resource languages, as there is also a challenge in finding material for those languages.

---

### Official Review · Reviewer_MCXi · 2024-11-04

**Soundness:** 2
**Presentation:** 3
**Contribution:** 2
**Rating:** 5
**Confidence:** 4

**Summary:**

This paper presents an evaluation of Aya, a multilingual language model, on four tasks including Aspect-Based Sentiment Analysis, Hate Speech Detection, Irony Detection, and Question-Answering, highlighting its strengths and limitations, particularly when compared to transformer-based models for the Portuguese language.

**Strengths:**

- This paper is centered around a very interesting and highly relevant topic.

- Clearly scoped tasks and objectives.

**Weaknesses:**

There are a few points that must be addressed:

Throughout the text it feels like the authors use aggressive speech, hate speech, and offensive speech interchangeably. For example, in Figure 1 we see offensive vs non-offensive. The authors should pay attention to this aspect: clearly define the specific type of abusive phenomena they are focusing on, and use that terminology consistently throughout their work – please look into the nuances surrounding the overlapping abusive phenomena (the distinction between hate speech, abusive language, and offensive language). See for example the work of Poletto et al.:

*Poletto F, Basile V, Sanguinetti M, Bosco C, Patti V. Resources and benchmark corpora for hate speech detection: a systematic review. Language Resources and Evaluation. 2021 Jun;55:477-523.*

I would have liked for the authors to spend a little bit more space detailing the methodology. The authors should provide the criteria used for selecting the examples, as well as the exact prompts used for all the tasks (not just QA):

-  How did the authors ensure that the examples were representative and diverse?
- What was the exact input for the Aya model? The authors provide the prompt for the QA, but not for the other tasks. Does that mean that for the other, only the examples were used? I am asking this because I am surprised by the fact that for hate speech *‘the generation was more efficient when using the labels as numbers, instead of the actual labels’* (cf. lines 309-311). For example, I just asked Aya if it is familiar with the hate speech definition provided by the OHCHR, and the answer was positive. How would the generation have changed if including this type of information? Did the authors provide any task-specific instructions to the model beyond the few-shot examples?

A more in-depth error analysis would have been interesting to have. The authors could consider a subset of the misclassified data and construct aggregate statistics over modes of failure which would then inform us how prevalent each of the kinds of mistake are. This would be useful for future research, as it would become possible to prioritize on which kind of misclassification to work on next.
In regards to the ABSA example provided, I don’t agree that *‘hotel’* has a neutral sentiment – it seems to be conflict (i.e., both positive and negative) or, we could say that the entity hotel has a positive sentiment towards the attributes location and service, but negative towards the attribute that would incorporate size/room description.

Was there any hyperparameter tuning performed for the transformer-based models? Interesting results for the Albertina models on the ID task.

Suggestions:

- abbreviations should be presented once, at their first occurrence, and always used afterwards (except for the abstract)
- I believe the paragraph starting on line 089 is actually a continuation of the previous one and does not require splitting
- line 124: an -> a
- line 161: a -> an

**Questions:**

Please see the above.

---

> ### Author Response · Authors · 2024-11-27
>
> **Weaknesses:**
>
> **1:** We do not currently have specific datasets, up until this point, for each type of offensive speech in the Portuguese language. Therefore, we are using a dataset that contains toxic speech, which considers hate speech, offensive speech, and aggressive speech as the same category.
>
> **2:** Regarding the methodology:
>
> - As presented in the article, we aimed to balance the examples by considering the labels and adhering to the token limit allowed by the model as input. Thus, for ID, for instance, we would have 50% of the examples as ironic and 50% as non-ironic.
>
> - We edited the file and have now made the examples available in the appendices. Regarding the question about the prompt used in the model, we only used few-shot examples as prompt for the HS. While for QA, ID, and ABSA tasks, we needed to include specific instructions in the prompt to ensure the response format. We also added this information now. We did not encounter any issues with the HS task using a more simplified prompt, although performance was not evaluated considering different prompts.
>
> **3:**
>
> - Regarding the error analysis: We agree that an in-depth error analysis could provide more information about the nature of the misclassifications. In future work, we plan to explore a subset of the misclassified data and perform a more detailed analysis of the failure modes.
>
> - In regards to the ABSA example: In this case, the way the dataset is annotated, ‘hotel’ is an aspect of that entity, as if it were a general aspect, which was annotated as “neutral”. The other aspects (for example, breakfast) are also included as aspects in the example, but with other polarities, which is why they were not explicitly mentioned in the text. Also, this specific dataset (one of the few available for PT-BR) does not consider “conflict” as a possible polarity, only positive, negative and neutral.
>
> - _Was there any hyperparameter tuning performed for the transformer-based models?_ Yes, indeed. The hyperparameters are available in another one of our papers, which we cite in the related works, but we have concealed them due to the anonymous review process. The configuration was as follows: The experiments used two Albertina models (Base and Large) with different hyperparameter settings. The Base model was configured with 12 attention heads, a batch size of 8, 3 training epochs, hidden layer size of 768, 12 hidden layers, a learning rate of 1e-5, CrossEntropy loss function, and the AdamW optimizer. On the other hand, the Large model had 16 attention heads, a batch size of 2, also with 3 training epochs, hidden layer size of 1536, 24 hidden layers, a learning rate of 1e-5, CrossEntropy loss function, and the AdamW optimizer. It is important to note that, in the QA experiment, there was an exception regarding the batch size, with a value of 16 used for the Base model and 8 for the Large model due to computational memory constraints.
>
> **4:** Thank you for your valuable suggestions. We fixed them in the paper.

---

### Official Review · Reviewer_opCB · 2024-11-12

**Soundness:** 2
**Presentation:** 3
**Contribution:** 1
**Rating:** 1
**Confidence:** 4

**Summary:**

This paper evaluates the performance of the multilingual Aya model across tasks such as Aspect-Based Sentiment Analysis (ABSA), Hate Speech Detection (HS), Irony Detection (ID), and Question-Answering (QA) in Brazilian Portuguese. Through a few-shot learning approach, Aya demonstrates competitive results in QA, surpassing some Portuguese-specific models, though it underperforms in tasks involving nuanced or slang-heavy language like HS. The study highlights Aya's potential in low-resource contexts while indicating the need for further tuning for certain language-specific tasks to match or exceed specialized models​

**Strengths:**

1. The paper is well-structured with a clear methodology.

2. It offers some insights into Aya's performance in multilingual contexts and addresses challenges faced by low-resource languages.

**Weaknesses:**

1. This paper only conducts an evaluation of the multilingual large language model on a specific language, Brazilian Portuguese, even though Aya supports 101 languages. The tasks are limited to Aspect-Based Sentiment Analysis (ABSA), Hate Speech Detection (HS), Irony Detection (ID), and Question-Answering (QA). Many other NLP tasks could be studied, such as reading comprehension, syntax parsing, named entity recognition, and event extraction. The evaluation scope and language focus are limited, reducing the paper's contribution.

2. Describing the equations for precision and recall in such detail seems unnecessary and only increases the document length without adding value.

3. This paper lacks inspiring conclusions from the experiments. It only presents main results and a confusion matrix, without providing in-depth analysis through fine-grained evaluation or insights into the working principles of the Aya model.

**Questions:**

See Above

---

> ### Author Response · Authors · 2024-11-27
>
> **Weaknesses:**
>
> **1:** The group has other works related to these tasks in the Portuguese language, with one of the objectives being to compare Aya's results with those obtained from other models that support the Portuguese language. We also aim to encourage the study of Portuguese and its exploration in models that support it, to increase the amount of available resources. Furthermore, as mentioned in the article, Portuguese is a low-resource language, and therefore, not all NLP tasks have a dedicated dataset available for it.
>
> **3:** We evaluate the model more generally according to the task, and justify the results according to those obtained. But the comment you left is really very important, and we added new examples to the work in order to better demonstrate the results.
>
> Thank you for your feedback.

---

### Author Response · Authors · 2024-11-27

**Dear reviewers,**

We would like to express our sincere gratitude for your valuable suggestions, comments, questions, and constructive criticisms. Your feedback has been incredibly helpful in improving the quality and clarity of our research. The paper will be updated shortly with the following changes:

- We will clarify that the dataset used for the Hate Speech Detection task contains toxic speech, categorizing hate speech, offensive speech, and aggressive speech under the same category.
- The appendices will include the few-shot examples.
- We will improve the text to clarify that this research explores the results of Aya, while prior work is related to all other models.
- We will comment on why neutral examples are harder to classify than positive and negative ones in the ABSA task, as ambiguity is more common in neutral examples.
- We will make other modifications related to grammar.

Once again, we thank you for your time and expertise.

---

> ### Author Response · Authors · 2024-11-28
> **Updated!**
>
> The paper is now updated.
>
> Thank you.

---

### Meta-Review · Area_Chair_rVwB · 2024-12-19

**Metareview:**

The paper evaluates Aya’s performance on four tasks: Aspect-Based Sentiment Analysis, Hate Speech Detection, Irony Detection, and question answering in the low-resource Portuguese language. The results show that while Aya performs well in certain tasks like Question-Answering, it struggles in others. The problem investigated in this paper is important. However, the paper is poorly written, and the reviewers raised many writing issues, which need to be addressed by significantly revising the paper. In addition, the paper lacks in-depth error analysis and inspiring conclusions from the experiments, and the novelty of the work is very limited.

**Additional Comments On Reviewer Discussion:**

One reviewer increased the score from 3 to 5 during the rebuttal period and no one strongly supported this paper.

---

### Decision · Program_Chairs · 2025-01-22

Reject